# Hospital management of major stroke types in Chinese adults: a population-based study of 20 000 hospitalised stroke cases

Haiqiang Qin ![ORCID],[1,2] Iain Turnbull,[3] Yiping Chen,[3,4] Neil Wright ![ORCID],[3] Liping Liu,[1,2] Pei Pei,[5] Wei Tang,[6] Shengping Xiang,[7] Yu Guo,[5] Xingquan Zhao,[1,2] Robert Clarke ![ORCID],[3] Liming Li,[5,8] Yongjun Wang,[1,2] Zhengming Chen,[3] On behalf of The China Kadoorie Biobank Collaborative Group

HQ, IT and YC contributed equally.

For numbered affiliations see end of article.

**Correspondence to**
Dr Robert Clarke;
robert.clarke@ndph.ox.ac.uk

## ABSTRACT

**Objectives** To compare hospital treatments for major stroke types in Chinese adults by stroke pathological types, sex, age, calendar year, hospital tier, region and other factors.

**Design** Cross-sectional analysis of medical records retrieved from 20 229 stroke cases in the China Kadoorie Biobank.

**Setting** Ten diverse areas (five urban, five rural) in China.

**Participants** First-incident stroke cases who were recruited during an 11-year follow-up of 0.5M participants in the China Kadoorie Biobank.

**Methods** Electronic copies of medical records of stroke cases were retrieved for clinical adjudication by local neurologists. Stroke cases were classified as ischaemic stroke (IS) (including lacunar infarction (LACI) and non-LACI (non-LACI)), intracerebral haemorrhage (ICH), subarachnoid haemorrhage (SAH) and unspecified stroke types.

**Results** Among 20 299 first-ever stroke cases, 17 306 (85%) had IS, 7123 had non-LACI, 6690 had LACI, 3493 had silent LACI, 2623 (13%) had ICH and 370 (2%) had SAH. Among IS cases, antiplatelet treatment was used by 64% (65% non-LACI, 66% LACI, 56% silent LACI), lipid-lowering by 50% (52% non-LACI, 53% LACI, 43% silent LACI) and blood pressure-lowering by ~42% of all IS types, with positive trends in the use of these treatments by calendar year and hospital tier. Among ICH cases, 53% used blood pressure-lowering and 10% used lipid-lowering treatments, respectively. In contrast, traditional Chinese medicines (TCMs) were used by 59% of IS (50% non-LACI, 62% LACI, 74% silent LACI), 38% of ICH and 30% of SAH cases, with positive trends by calendar year and by hospital tier.

**Conclusions** Among IS cases, use of antiplatelet and lipid-lowering medications increased in recent years, but use of TCM still exceeded use of blood pressure-lowering treatment. In contrast, blood pressure-lowering treatment was widely used for ICH, but only half of all ICH cases used blood pressure-lowering treatment.

## INTRODUCTION

Stroke is a major cause of death and disability worldwide, with three-quarters (~6.3 million) of deaths each year occurring in developing countries including China.[1–3] In China, stroke is now the leading cause of death and disability, accounting for almost 2 million deaths in 2017.[3–5] While mortality rates from stroke have declined substantially over the last few decades in high-income countries, the annual costs for hospital treatment and rehabilitation of stroke were ¥40 billion and approximately US$6 billion in 2016, which were 10-fold higher than those for ischaemic heart disease (IHD).[3] The average per-patient cost of hospitalisation for stroke cases in China in 2018 was ¥14 29 (¥9409 (US$1309) for ischaemic stroke (IS) and ¥19 149 (US$2772) for haemorrhagic stroke).

In recent decades, evidence-based guidelines for treatment of different stroke pathological types have been developed, regularly updated and widely disseminated in China. In contrast, whereas traditional Chinese medicines (TCMs), for which no randomised evidence exists, continue to be extensively used for the treatment of hospitalised stroke cases.[6] Nevertheless, there is still substantial

variation in access to and use of evidence-based treatments for stroke in different regions of China.[6] While such treatments have been extensively studied in Western countries,[7–9] little is known about the contemporary use of evidence-based medical and surgical treatments for hospitalised cases of stroke in China. Previous studies of hospital management of stroke in China have been constrained by small sample sizes and involvement of only large teaching or tertiary hospitals, or restricted to studies in large urban areas only.[10–15]

### Strengths and limitations of this study

► The China Kadoorie Biobank is a nationwide community-based prospective cohort study involving over 0.5 million men and women from five rural and five urban areas in China.
► Local neurologists classified 20 000 first-incident stroke cases that were reported over an 11-year follow-up into relevant pathological types.
► The data collection on use of medication was limited to in-hospital treatments only.
► The study was not representative of the Chinese population and data collection was limited to first-ever stroke cases.

We examined the use of medical and surgical treatments for first-ever incident stroke cases by pathological type, overall and by sex, age of onset, calendar year, hospital tier, region and other risk factors during an 11-year follow-up of 0.5 million adults in the China Kadoorie Biobank (CKB).[16 17]

## METHODS

### Study population

Details of the study design and methods for recruitment and follow-up used in the CKB have been previously reported.[16 17] In brief, the 2004–2008 baseline survey was conducted in five urban cities (Qingdao, Harbin, Liuzhou, Haikou and Suzhou) and five rural counties (Henan, Hunan, Sichuan, Gansu and Zhejiang) in China, selected to maximise regional differences in risk exposures and prevalence of major diseases. In each area, all permanent residents aged 35–74 years who were without disability were invited to participate. A total of 512 726 individuals (including some outside the target age range) agreed to participate in the study. The main analyses of the present report were restricted to adjudicated first-ever incident cases of stroke occurring prior to 31 December 2017 among 484 413 participants with no self-reported heart disease or stroke/transient ischaemic attack at baseline (n=23 129) or stroke prior to their first adjudicated stroke (n=5184).

Data on sociodemographic status, smoking and alcohol consumption, diet, physical activity and medical history were collected by trained health workers using laptop-based questionnaires at local study clinics. All participants also had clinical measurements recorded, including height, weight, waist and hip circumference, lung function, blood pressure and heart rate and a blood sample collected. All participants provided written informed consent for participating and for accessing their medical records during follow-up by study investigators.

After the baseline survey, the vital status of participants was monitored regularly through Disease Surveillance Points death registries, supplemented by annual checks with local residential records and active confirmation through local street committees or village administrators.[17] The occurrence of stroke among participants was identified by electronic linkage, using a unique personal identification number, to the national health insurance (HI) system that covered >97% of CKB participants, and

to death and disease-specific (stroke, IHD, cancer and diabetes) registries accessible in the local study areas.

Selected medical records of hospitalised cases of first-ever reported stroke cases were retrieved and the diagnosis was verified by trained local CKB staff at each of the 261 different hospitals. The retrieved documents (eg, initial covering page and discharge summary of the first admission record, with reports of brain imaging and use of prespecified medical treatments for stroke) were reviewed electronically by 26 accredited Chinese neurologists and stroke physicians from tier III (top ranking) hospitals in China using bespoke IT systems. While indicators of medical treatments were pre-specified at three time periods (first 24 hours after admission, during hospital stay, and at discharge) in the medical notes, information on timing of such treatments was incomplete and consequently it was not possible to conduct separate analysis of use of drug treatment during each time period. Hence, recorded hospital treatment in any of the three periods was used in the present analysis.

### Classification of stroke pathological types

All reported and adjudicated diagnoses of stroke pathological types were checked centrally by study staff and standardised by 10th Revision of the International Statistical Classification of Diseases and Related Health Problems (ICD-10) codes using bespoke software. The major stroke pathological types examined were IS (ICD-10 I63, including lacunar infarction (LACI) and non-LACI (non-LACI)), intracerebral haemorrhage (ICH; I61), subarachnoid haemorrhage (SAH; I60) and unspecified stroke (I64). IS was defined as focal neurological dysfunction lasting for more than 24 hours with or without neuroimaging evidence of a cerebral infarct. LACI was classified by the evidence of an infarct <1.5 cm in diameter on CT or MR brain imaging with focal neurological deficits, including lacunar syndrome.[18] Silent lacunar infarct silent LACI was characterised by imaging evidence of a cerebral infarct, 1.5 cm in diameter in the absence of any focal neurological deficit. Non-LACI was defined as any IS type excluding LACI.

ICH was defined as a neurological dysfunction caused by haemorrhage into the brain parenchyma or the ventricular system, excluding those induced by injury, with or without neuroimaging evidence of haemorrhage. SAH was defined as a neurological dysfunction caused by a haemorrhage into the subarachnoid space, excluding those induced by injury, with or without relevant neuroimaging evidence. In addition, all silent LACI cases were reviewed centrally by two research clinicians with relevant training in neurology and included with other stroke cases in an 'any stroke' endpoint.

### Statistical analysis

The present study included a total of 20 299 first ever incident stroke cases with no prior history of a CVD event at baseline questionnaire or stroke event before the first adjudicated event. All cases were adjudicated online by

**Table 1** Selected baseline characteristics of participants by types and subtypes of first adjudicated stroke

| | IS | | | ICH | SAH | Any stroke | All CKB |
|---|---|---|---|---|---|---|---|
| | **Non-LACI** | **LACI** | **Silent LACI** | | | | |
| | **(n=7123)** | **(n=6690)** | **(n=3493)** | **(n=2623)** | **(n=370)** | **(n=20 299)** | **(n=484 413)** |
| **Age, %** | | | | | | | |
| 30–39 | 2.0 | 1.8 | 2.0 | 4.4 | 6.3 | 2.4 | 16.0 |
| 40–49 | 13.1 | 14.3 | 13.4 | 16.9 | 21.4 | 14.5 | 31.0 |
| 50–59 | 32.5 | 34.3 | 33.0 | 34.6 | 36.6 | 33.8 | 30.8 |
| 60–69 | 35.5 | 34.4 | 35.1 | 30.0 | 24.8 | 33.9 | 16.6 |
| 70–79 | 16.9 | 15.2 | 16.4 | 14.2 | 11.0 | 15.5 | 5.7 |
| Mean (SD) | 60.1 (9.6) | 59.6 (9.5) | 60.0 (9.6) | 58.3 (10.0) | 56.5 (10.2) | 59.5 (9.7) | 51.5 (10.5) |
| **Sex, %** | | | | | | | |
| Male | 50.0 | 46.5 | 39.0 | 53.1 | 36.9 | 47.4 | 40.8 |
| Female | 50.0 | 52.3 | 59.8 | 46.9 | 60.3 | 52.6 | 59.2 |
| **Area, %** | | | | | | | |
| Rural | 58.0 | 40.6 | 39.0 | 63.3 | 51.6 | 49.2 | 56.8 |
| Urban | 42.0 | 59.4 | 61.0 | 36.7 | 48.4 | 50.8 | 43.2 |
| **Highest education, %** | | | | | | | |
| No formal/ primary school | 54.4 | 52.4 | 53.2 | 54.8 | 46.4 | 53.4 | 50.7 |
| Middle/high school | 40.1 | 42.4 | 38.8 | 40.3 | 40.1 | 42.3 | 43.6 |
| College/ university | 4.0 | 4.1 | 4.8 | 4.1 | 5.6 | 4.3 | 5.7 |
| **Annual household income (RMB), %** | | | | | | | |
| <RMB9999 | 30.1 | 27.3 | 29.6 | 30.6 | 29.6 | 29.2 | 28.3 |
| RMB10 000–RMB19 999 | 29.4 | 33.5 | 30.1 | 30.6 | 24.0 | 30.7 | 28.8 |
| RMB20 000–RMB34 999 | 22.9 | 21.0 | 21.4 | 24.3 | 23.5 | 23.6 | 24.7 |
| ≥RMB35 000 | 16.0 | 17.0 | 15.9 | 13.7 | 15.1 | 16.5 | 18.1 |
| **Current smoker, %** | | | | | | | |
| Males | 69.4 | 67.1 | 55.1 | 62.5 | 52.1 | 66.5 | 62.0 |
| Female | 3.5 | 3.6 | 1.8 | 2.2 | 4.3 | 3.0 | 2.3 |
| **Current alcohol drinker, %** | | | | | | | |
| Males | 35.2 | 38.6 | 33.9 | 38.8 | 24.8 | 36.5 | 33.9 |
| Female | 2.5 | 1.4 | 2.0 | 3.7 | 1.3 | 2.5 | 2.1 |
| **Regular dietary intake*, %** | | | | | | | |
| Meat | 79.2 | 82.0 | 79.1 | 81.7 | 72.3 | 82.3 | 82.9 |
| Fish | 44.8 | 45.4 | 42.4 | 42.8 | 41.6 | 45.6 | 46.8 |
| Dairy | 16.8 | 18.6 | 19.5 | 16.2 | 19.1 | 18.3 | 19.7 |
| Fruit | 55.1 | 54.8 | 54.9 | 54.8 | 58.6 | 56.3 | 59.5 |
| **Prevalent disease†, %** | | | | | | | |
| Diabetes | 6.8 | 5.9 | 4.1 | 3.2 | 2.0 | 5.3 | 2.7 |

Continued

**Table 1** Continued

| | IS | | | | | | |
| | Non-LACI | LACI | Silent LACI | ICH | SAH | Any stroke | All CKB |
| | (n=7123) | (n=6690) | (n=3493) | (n=2623) | (n=370) | (n=20299) | (n=484413) |
|---|---|---|---|---|---|---|---|
| Hypertension | 21.8 | 19.3 | 19.3 | 23.5 | 14.3 | 20.8 | 9.7 |
| Clinical measures, mean (SD) | | | | | | | |
| SBP (mm Hg) | 140.3 (23.9) | 138.4 (23.9) | 132.8 (22.6) | 149.0 (24.5) | 124.7 (22.6) | 142.1 (24.2) | 130.4 (20.9) |
| BMI (kg/m$^2$) | 24.3 (3.6) | 24.0 (3.4) | 23.3 (3.4) | 24.0 (3.6) | 21.7 (3.5) | 24.4 (3.5) | 23.6 (3.4) |
| RPG (mmol/L) | 6.8 (3.5) | 6.4 (3.3) | 6.1 (2.9) | 6.2 (2.9) | 5.7 (3.2) | 6.5 (3.2) | 6.0 (2.3) |
| Hospital tier, % | | | | | | | |
| Tier 1 (162 hospitals)‡ * | 2.7 | 4.5 | 2.6 | 2.6 | 3.4 | 3.2 | |
| Tier 2 (53 hospitals) | 4.2 | 6.2 | 3.7 | 4.7 | 3.1 | 4.7 | |
| Tier 3 (46 hospitals) | 91.6 | 88.1 | 90.6 | 91.9 | 85.7 | 92.0 | |

Means and percentages are directly standardised to age, sex and study area structure of the CKB study population after excluding 23129 participants with prior CHD or stroke/TIA and 5184 participants with a reported stroke prior to first adjudicated stroke, as appropriate.
*1–3 days per week or more often.
†Self-reported at baseline survey.
‡Includes 126 community health centres.
BMI, body mass index; CHD, coronary heart disease; CKB, China Kadoorie Biobank; ICH, intracerebral haemorrhage; LACI, lacunar infarction; RPG, random plasma glucose; SAH, subarachnoid haemorrhage; SBP, systolic blood pressure; ; TIA, transient ischaemic attack.

senior neurologists in China and reviewed centrally by study clinicians in Oxford. The rates of incident stroke cases were standardised to the overall CKB population by sex, area and age at diagnosis. Rates of use of medical and surgical treatments were estimated separately for any stroke and for stroke pathological types, by sex and area (rural vs urban), separately. All analyses were conducted using R V.3.5.0.

## Patient and public involvement

Prior to recruitment in the CKB, local community leaders in China were consulted. The CKB study findings[19] are routinely reported in peer-review publications and any relevant public health messages disseminated[20] using local press, television and internet.

## RESULTS
### Characteristics of cases with different stroke types

Among 484413 participants included in the present analysis, the mean age (SD) at baseline was 51.5 (10.5) years, 59% were women, and 43% lived in urban areas (table 1). During a median follow-up of 11 years (4.72 million person-years), a total of 61902 incident stroke cases were reported, including 57190 (92%) hospitalised cases. Overall, there were 20299 first-ever hospitalised cases of primary stroke for which each diagnosis was confirmed by clinical adjudication, including 17306 (85%) with IS (7123 non-LACI, 6690 LACI and 3493 silent LACI), 2623 with ICH and 370 with SAH, respectively. Overall, 93% of

the reported stroke cases (92% for IS, 97% for ICH and 96% for SAH) had brain imaging.

Compared with those without stroke, individuals who experienced first-ever stroke during follow-up were older and more likely to be male, to reside in urban areas, and to have lower levels of education and household income at baseline (table 1). Individuals with first-ever ICH were more likely than those with other stroke types to be men, to live in rural areas, and to have lower levels of full-time education (54.8% had no formal/primary schooling: table 1) and have low annual household income (30.6% had income less than ¥10 000: table 1). Likewise, individuals with ICH had the highest proportion with self-reported hypertension (23.5%: table 1), and the highest mean levels of systolic blood pressure (SBP) (149 mm Hg: table 1). With the exception of LACI cases (who were more likely to be female and live in urban areas), individuals with non-LACI and LACI had broadly similar baseline characteristics (table 1). In addition, the characteristics of individuals with LACI and silent LACI cases were comparable. However, compared with LACI, silent LACI cases had lower mean levels of SBP (133 mm Hg vs 138 mm Hg). The baseline characteristics of individuals with SAH were generally similar to those with ICH (table 1). Interestingly, the overall prevalence of atrial fibrillation was very low, although higher in non-LACI than in LACI cases (0.3% vs 0.1%: table 2).

**Table 2** Hospital managements for subtypes of IS

| Treatment, n (%) | Non-LACI (n=7123) | | LACI (n=6690) | | Silent LACI (n=3492) | | All IS (n=17 305) | | P value |
|---|---|---|---|---|---|---|---|---|---|
| Any antiplatelet agent | 4638 | (65.1) | 4434 | (66.3) | 1943 | (55.6) | 11 015 | (63.7) | <0.001 |
| Aspirin | 3829 | (53.8) | 3722 | (55.6) | 1761 | (50.4) | 9312 | (53.8) | <0.001 |
| Clopidogrel | 456 | (6.4) | 431 | (6.4) | 132 | (3.8) | 1019 | (5.9) | <0.001 |
| Dual antiplatelet therapy | 589 | (8.3) | 413 | (6.2) | 90 | (2.6) | 1092 | (6.3) | <0.001 |
| Anticoagulant agent | 689 | (9.7) | 569 | (8.5) | 146 | (4.2) | 1404 | (8.1) | <0.001 |
| Any antihypertensive agent | 2976 | (41.8) | 2820 | (42.2) | 1426 | (40.8) | 7222 | (41.7) | 0.439 |
| 1 antihypertensive agent | 2217 | (31.1) | 2137 | (31.9) | 1046 | (30.0) | 5400 | (31.2) | 0.119 |
| 2 antihypertensive agent | 746 | (10.5) | 668 | (10.0) | 354 | (10.1) | 1768 | (10.2) | 0.629 |
| ≥3 antihypertensive agent | 170 | (2.4) | 156 | (2.3) | 110 | (3.2) | 436 | (2.5) | 0.028 |
| Any lipid-lowering agent | 3670 | (51.5) | 3522 | (52.6) | 1508 | (43.2) | 8700 | (50.3) | <0.001 |
| Statin | 3601 | (50.6) | 3441 | (51.4) | 1461 | (41.8) | 8503 | (49.1) | <0.001 |
| Other lipid lowering agent | 133 | (1.9) | 130 | (1.9) | 63 | (1.8) | 326 | (1.9) | 0.879 |
| Thrombolytic therapy | | | | | | | | | <0.001 |
| r-tPA | 25 | (0.4) | 5 | (0.1) | 0 | (0.0) | 30 | (0.2) | <0.001 |
| Urokinase | 60 | (0.8) | 15 | (0.2) | 0 | (0.0) | 75 | (0.4) | <0.001 |
| Defibrase | 192 | (2.7) | 179 | (2.7) | 43 | (1.2) | 414 | (2.4) | <0.001 |
| Any TCM | 3545 | (49.8) | 4151 | (62.0) | 2571 | (73.6) | 10 267 | (59.3) | 0.095 |
| Atrial fibrillation | 24 | (0.3) | 8 | (0.1) | 4 | (0.1) | 36 | (0.2) | <0.001 |

IS, ischaemic stroke; LACI, lacunar infarction; TCM, traditional Chinese medicines.

## Hospital treatment in is cases

Overall, antiplatelet agents (aspirin or clopidogrel) were administered to 63.7% of all hospitalised IS cases, but use of antiplatelet agents was higher in cases with LACI (66.3%) and non-LACI (65.1%) than silent LACI cases (55.6%) (table 2). Dual antiplatelet therapy was much less commonly used (non-LACI 8.3%, LACI 6.2% and silent LACI 2.6%, respectively). Intravenous thrombolytic treatments (Recombinant Tissue Plasminogen Activator (rtPA) and urokinase) were used in 1.2% of non-LACI and 0.3% of LACI cases. Overall, 42% of all IS cases received blood pressure-lowering treatments, with similar proportions for different IS types (42.2%, 41.8% and 40.8% for LACI, non-LACI and silent LACI, respectively). Likewise, lipid-lowering treatments were used by only about half of all hospitalised cases of IS (52.6%, 51.5% and 43.2% for LACI, non-LACI and silent LACI, respectively). In contrast, 56% of IS cases (62.0%, 49.9%, and 73.6%, for LACI, non-LACI and silent LACI, respectively) received TCM (table 2).

For antiplatelet and lipid-lowering medications in addition to TCM, the rates of use for IS increased with age, while no such increasing trends with age were observed for use of blood-pressure lowering treatments (figure 1). All medical treatments for IS, with the exception of blood pressure-lowering therapy, increased by calendar year of follow-up between 2005 and 2017 (figure 2). While the use of blood pressure-lowering treatments did not differ significantly by hospital tiers, the use of TCM was higher in tier III hospitals and increased by calendar year between 2005 and 2017 (figure 2). Similarly, age-specific use of antiplatelet, blood pressure-lowering and lipid-lowering agents and TCM among IS cases did not vary appreciably by sex or area of residence (online supplemental efigures 1–6), but use of TCM increased by hospital tier in both men and women and in rural and urban areas. Carotid endarterectomy and stenting were very rarely undertaken (table 2).

## Hospital management of ICH and SAH

Among participants hospitalised with ICH, 3.9% had surgical clot evacuation. A small proportion of cases with SAH underwent endovascular coiling (7.0%) or neurosurgical clipping of an aneurysm (5.7%). Approximately half of ICH (52.6%) or SAH (50.5%) cases received blood pressure-lowering medication, which did not vary significantly by sex or area (online supplemental efigure 1). Overall, the use of lipid-lowering agents was 10.4% for ICH whereas approximately one-third of ICH (37.8%) and SAH (29.7%) cases received TCM while in hospital (table 3).

## DISCUSSION

The present study, involving over 20 000 incident stroke cases, demonstrated that while there have been substantial improvements in access to and use of brain imaging for diagnosis of stroke types in hospitalised cases, the use

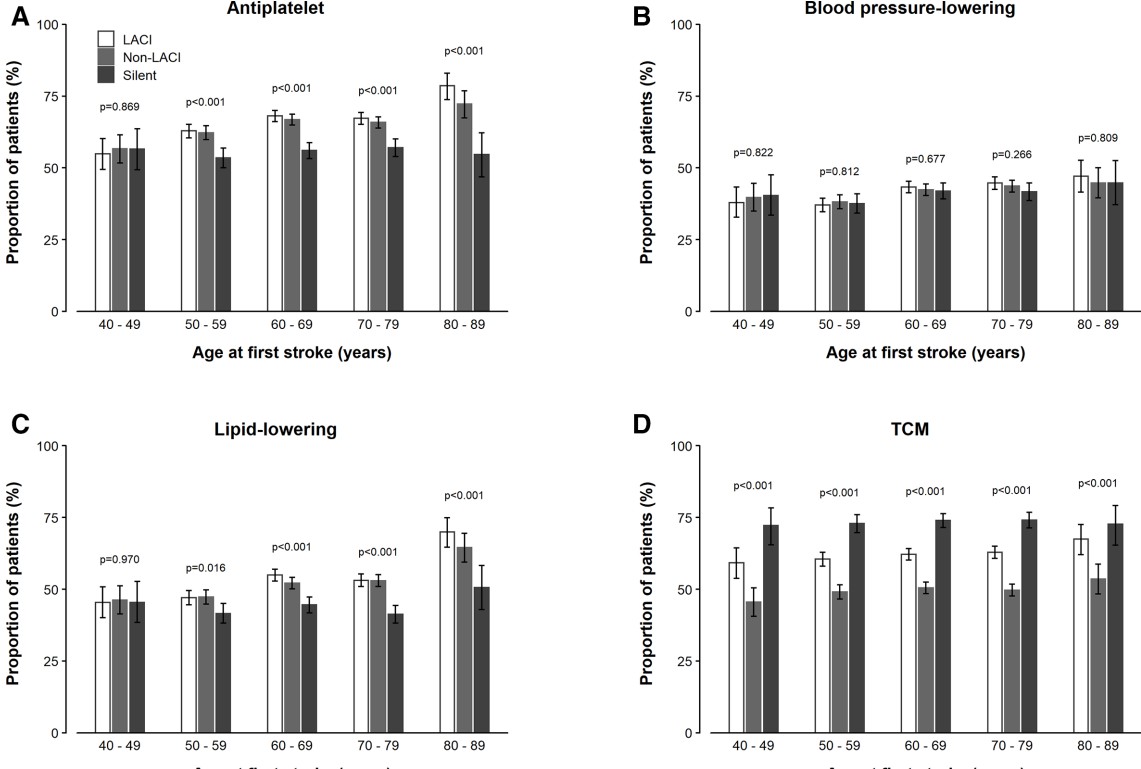

**Figure 1** Proportions of medical treatments in hospital by is subtypes. LACI, lacunar infarction; TCM, traditional Chinese medicines.

of many evidence-based medical treatments, including antiplatelet, blood pressure-lowering and lipid-lowering agents, remains low in China, even in top-rank hospitals. In contrast, the use of TCM has remained consistently high, especially in higher-ranking hospitals, highlighting the urgent need for better utilisation of evidence-based treatments, particularly in IS cases. Likewise, further trials are required to examine the efficacy and safety of such treatments in ICH cases. Importantly, further assessments of the use of TCM in the management of both IS and ICH are also indicated.[19]

Despite marked improvements in the hospital management of stroke in recent decades in China,[15] the use of evidence-based medications remains lower than in Western countries. Thrombolytic therapy (eg, r-tPA), which was approved by the State Food and Drug Administration of China in 2001, was administered to <1% of potentially eligible cases in the present study, compared with 3.0%–8.5% reported in the USA[20] and 1.6% reported by the Chinese National Stroke Registry (CNSR) in 2011.[21]

A recent American Heart Association registry study conducted during the same calendar period reported a higher use of antiplatelet agents in the USA versus China (83.9% vs 64%, respectively).[22] Since data on common adverse events including bleeding were not recorded in the present study, the possibility that the lower use of antiplatelet therapy was clinically indicated in this population cannot be entirely excluded. Likewise, the use of

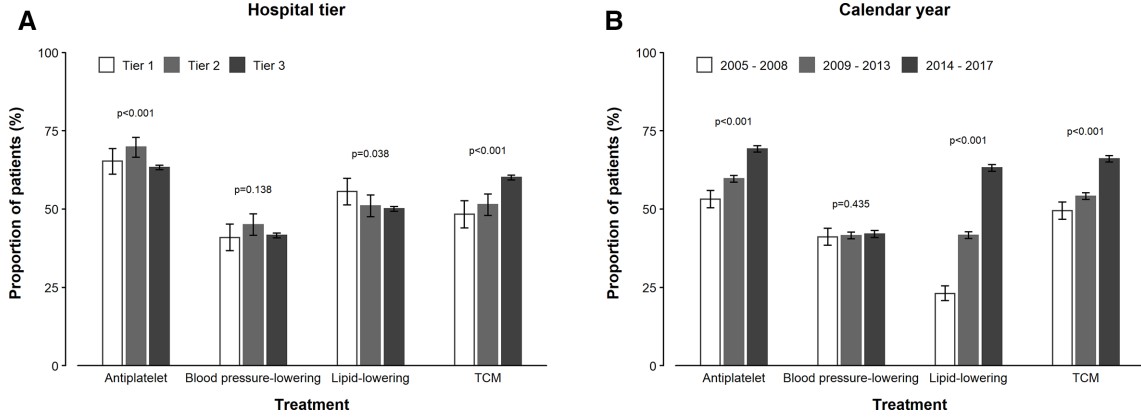

**Figure 2** Use of antiplatelet, antihypertensive, lipid-lowering and TCM treatments for ischaemic stroke by hospital tiers and calendar periods from 2005 to 2017. TCM, traditional Chinese medicines.

**Table 3** Hospital management for ICH and SAH

| Hospital management, n (%) | ICH (n=2623) | | SAH (n=370) | |
|---|---|---|---|---|
| Brain imaging (CT or MRI) | | (97.0) | | (96.0) |
| Any antihypertensive agent | 1379 | (52.6) | 187 | (50.5) |
| 1 antihypertensive agent | 899 | (34.3) | 146 | (39.5) |
| 2 antihypertensive agent | 436 | (16.6) | 42 | (11.4) |
| ≥3 antihypertensive agent | 151 | (5.8) | 8 | (2.2) |
| Any lipid-lowering agent | 272 | (10.4) | 12 | (3.2) |
| Statin | 266 | (10.1) | 12 | (3.2) |
| Other lipid lowering agent | 8 | (0.3) | 0 | (0.0) |
| Any TCM | 992 | (37.8) | 110 | (29.7) |
| Surgery | 230 | (8.8) | 47 | (12.7) |
| Clot evacuation | 103 | (3.9) | 3 | (0.8) |
| Coiling | 0 | (0.0) | 26 | (7.0) |
| Aneurysm clipping surgery | 0 | (0.0) | 21 | (5.7) |

ICH, intracerebral haemorrhage; SAH, intracerebral haemorrhage; TCM, traditional Chinese medicines.

blood pressure-lowering and lipid-lowering medications was also higher in the USA compared with China (76.7% vs 41.7% and 77.6% vs 50.3%, respectively).[22] Moreover, the use of drug treatments for secondary prevention of stroke in the CKB was significantly lower than those reported in the CNSR.[15] Lowering blood pressure is the most important treatment to improve outcomes of cases with SAH and ICH.[23 24] However, in the present study, only about half of all ICH and SAH cases (and only 42% of IS cases) received blood pressure-lowering treatment. One explanation for the discrepant findings on rates of use of treatment in stroke cases in the CKB and the CNSR may be that the former study was conducted in 10 diverse regions (5 rural regions and 5 urban) with 21% of hospitals involving lower-grade (tier 1) hospitals, whereas recruitment of stroke cases in the latter study was restricted to those living in urban regions and those in higher-tier hospitals. For the same reason, the use of intravenous thrombolytic therapy was <1% in the CKB vs 1.5% in CNSR. Likewise, the prevalence of reported atrial fibrillation was lower than previously reported estimates from population studies in China, and only 7% of stroke cases had Holter monitoring investigations. In addition, medication data in the present study were obtained from retrieved hospital admission records rather than at later periods of follow-up. Although in-hospital treatment data are reliable, intended use of medication following discharge from hospital may be underestimated.

While there is substantial underuse of evidence-based medical and surgical treatments in hospitalised stroke cases, TCMs are widely used for all stroke types in China, particularly in tier 3 hospitals, despite absent randomised evidence to support their efficacy and safety.[19 25] The overall rates of use of evidence-based medications were also lower among younger stroke cases consistent with previous reports.[26] The economic and social effects of stroke and its long-term sequelae in younger cases are considerable and these inequalities should be a priority for improvement. The suboptimal use of medical treatments may partly explain the high recurrence rates of IS recently reported in this study population.[27]

Rates of medical therapy did not vary appreciably among IS types (non-LACI, LACI and silent LACI). However, relevant clinical guidelines address only the management of symptomatic stroke cases,[27 28] since there have been no randomised controlled trials including participants with silent LACI. The high prevalence of silent LACI—and its strong association with subsequent risks of symptomatic stroke,[29–31] cognitive decline and dementia,[32–34] psychiatric disorders[35] and increased mortality,[36] highlight the need for new trials to assess the efficacy and safety of medical treatments in such cases.

The use of drug treatments has increased considerably in recent years, particularly after introduction of universal health coverage in 2009, reflecting the effects of health policy. A recent report of the temporal trends in hospital admissions for stroke in China demonstrated that admissions for first stroke increased by 3%–6% per year after 2009 following adjustment for age, other risk factors and incidence of other diseases.[37] These stroke admissions did not, however, vary appreciably between rural and urban areas or between hospital tiers, suggesting that low socioeconomic status, which is more common in rural areas, is unlikely to explain the suboptimal use of evidence-based treatments for stroke in China.

Further studies on the adverse effects of medications (eg, bleeding risks with antiplatelet agents or rhabdomyolysis in high-dose statin users) are also needed to improve our understanding of the reduced rates of use of evidence-based therapies for IS in Chinese populations. The consequences of their suboptimal use—and the relatively prevalent use of TCM—on short-term mortality and stroke recurrence following stroke also warrant further study.

This study had several strengths, including the use of bespoke web-based adjudication systems to determine the diagnosis and clinical characteristics of over 20 000 reported stroke cases of all pathological types including silent LACI. Moreover, it included data from tier I hospitals in rural areas that have not been included in previous studies. However, there were several limitations. First, the CKB study was not a nationally representative study and, hence, the results cannot be representative of all areas in China. Second, it is possible that some selected clinical information relating to the acute phase of stroke (eg, administration of thrombolysis or aspirin treatment within 48 hours of hospital admission) was recorded in the emergency room and omitted in the retrieved medical records. Not all hospitals recorded medications for intended treatment after discharge in the medical records, which may have led to underestimates of their actual use. Lastly, the present study did not collect records of side effects

of treatments (eg, bleeding associated with the use of antiplatelet agents). Consequently, the relatively low observed rates of use of medical treatment may reflect clinical discretion or caution. However, the possibility of missing clinical data was minimised by photographing all available relevant medical records and enlistment of specialist neurologists and stroke physicians from tier III hospitals to adjudicate the diagnoses of stroke types.

Overall, this contemporary study of 20 000 hospitalised stroke cases demonstrated that use of all evidence-based medical treatments has increased substantially after the introduction of the nationwide universal healthcare system in China. Despite recent improvements in stroke management, there is still considerable underuse of many evidence-based treatments and widespread overuse of TCM for stroke in the absence of evidence on their efficacy and safety. Future studies should investigate the causes and consequences of variations in stroke care in clinical practice in China.[38] Moreover, further large randomised trials are needed to evaluate the efficacy and safety of established evidence-based treatments for silent LACI.

**Author affiliations**
[1]Department of Neurology, Beijing Tiantian Hospital, Capital Medical University, Beijing, China
[2]China National Clinical Research Centre for Neurological Diseases, Beijing, China
[3]Clinical Trial Service Unit and Epidemiological Studies, NDPH, University of Oxford, Oxford, UK
[4]Medical Research Council Population Health Research Unit (PHRU), Nuffield Department of Population Health, University of Oxford, Oxford, UK
[5]Chinese Academy of Medical Sciences and Peking Union Medical College, Beijing, Beijing, China
[6]Emergency Department, Pengzhou Traditional Chinese Medical Hospital, Sichuan, China
[7]Pengzhou Tongyi Hospital, Pengzhou, Sichuan, China
[8]Department of Epidemiology and Biostatistics, School of Public Health, Peking University Health Science Centre, Beijing, China

**Collaborators** China Kadoorie Biobank Collaborative Group (Members of the CKB Collaborative Group are shown in online supplemental material).

**Contributors** HQ, IT, YC and RC designed the study. NW concluded the statistical analyses. HQ wrote the first draft and IT, YC and RC revised the report, tables and figures. LLiu and YW provided critical advice on the analyses and interpretation and YG, LLi and ZC supervised the overall conduct of the CKB study. All authors provided critical comments on the final report. RC and YC took responsibility for submission of the final version for publication. YC and RC are guarantors for this report, controlled access to the data and the decision to submit for publication.

**Funding** The CKB baseline survey and the first resurvey were supported by the Kadoorie Charitable Foundation in Hong Kong. The long-term follow-up has been supported by Wellcome grants to Oxford University (212946/Z/18/Z, 202922/Z/16/Z, 104085/Z/14/Z, 088158/Z/09/Z) and grants from the National Key Research and Development Program of China (2016YFC0900500, 2016YFC0900501, 2016YFC0900504, 2016YFC1303904) and from the National Natural Silent LAClence Foundation of China (91843302). The UK Medical Research Council (MC_UU_00017/1,MC_UU_12026/2 MC_U137686851), Cancer Research UK (C16077/A29186; C500/A16896) and the British Heart Foundation (CH/1996001/9454), provide core funding to the Clinical Trial Service Unit and Epidemiological Studies Unit at Oxford University for the project .

**Competing interests** None declared.

**Patient consent for publication** Not applicable.

**Ethics approval** Ethical approval for the China Kadoorie Biobank was obtained from the Oxford University Tropical Research Ethics Committee (OxTREC; (2005);

OxTREC Reference 025-04), and the Chinese Center for Disease Control and Prevention Ethical Review Committee (2004; Reference 005/2004).

**Provenance and peer review** Not commissioned; externally peer reviewed.

**Data availability statement** Data are available on reasonable request. The China Kadoorie Biobank (CKB) is a global resource for the investigation of lifestyle, environmental, biochemical and genetic factors as determinants of common diseases. The CKB study group is committed to making the cohort data available to the scientific community in China, the UK and worldwide to advance knowledge about the causes, prevention and treatment of disease. Detailed information on data that are currently available for open access users and how to apply for these are provided on the study website (http://www.ckbiobank.org/site/Data+Access). Researchers who are interested in obtaining the individual data from the China Kadoorie Biobank study that underlines this paper should contact ckbaccess@ndph.ox.ac.uk. A research proposal will be requested to ensure that analyses will be performed by bona fide researchers.

**ORCID iDs**
Haiqiang Qin http://orcid.org/0000-0003-4500-7508
Neil Wright http://orcid.org/0000-0002-3946-1870
Robert Clarke http://orcid.org/0000-0002-9802-8241

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
