## [Reviewer comments · BMJ Open]

ARTICLE DETAILS

TITLE (PROVISIONAL)	Hospital management of major stroke types in Chinese adults: a population-based study of 20,000 hospitalised stroke cases
AUTHORS	Qin, Haiqiang; Turnbull, Iain; Chen, Yiping; Wright, Neil; Liu, Liping; Pei, Pei; Tang, Wei; Xiang, Shengping; Guo, Yu; Zhao, Xingquan; Clarke, Robert; Li, Liming; Wang, Yongjun; Chen, Zhengming

VERSION 1 – REVIEW

REVIEWER	Li, Shujuan Capital Medical University, Beijing Chaoyang Hospital Department of Neurology
REVIEW RETURNED	31-Aug-2021

GENERAL COMMENTS	In this manuscript, using data from the China Kadoorie Biobank, the authors studied the current situation of hospital management and evidence-based treatments for stroke in Chinese hospitals. The results indicate that many evidence-based stroke therapies are still underused in China, while traditional Chinese medicine (TCM) has been widely used in the absence of sufficient efficacy and safety data. These results may provide evidence for stroke treatments and future studies. There are a few minor shortcomings to be ironed out before publication. 1. The time and sample size of this study have been described for many times in the article. Please pay attention to the consistency of the descriptions. 1) Page 3, lines 11-14: "The China Kadoorie Biobank recruited >0.5 million adults during 2004-08 from ten diverse areas and medical records were retrieved from 20,229 participants (n=261 hospitals) hospitalised with a first-ever incident stroke over an 11-year follow-up period." 2) Page 4, lines 8-9: "In a 12-year follow-up of 0.5 million Chinese adults recruited into the China Kadoorie Biobank in 2004-2008". 3) Page 6, line 24: "In brief, the 2004-8 baseline survey was conducted". 4) Page 12, lines 3-5: "increased by calendar year of follow-up between 2005 and 2017." 2. In the "Study population" section, the description of CKB is relatively detailed, but it is not clear about the population of the present study. It is suggested to add the first sentence of "Statistical analysis" (Page 9, lines 29-37) to the "Study population", and the sentence should be amended to avoid ambiguity. Due to the large sample size of this study, it is suggested to add a flow chart of participants.
--

	3. In this article, silent cerebral infarction (SCI) was included in the classification of IS. Therefore, the description of IS and non-LACI needs to be adjusted. 4. "Surgical treatments" was mentioned in the "Introduction", but they are only mentioned again in the results and Table 2, and these results are not discussed in the "Discussion" section. Therefore, I suggest that this part can be deleted as appropriate. 5. Supplement the description of "Unjustified stroke (N=26,965)" in the eTable 1. There is no description of this number in the manuscript or annotations.
--	--

REVIEWER	Ding, Jing
	Zhong Shan Hospital
REVIEW RETURNED	11-Sep-2021

GENERAL COMMENTS	The study aimed to investigate the use of hospital treatments for stroke types in Chinese adults. Over 0.5 Million Chinese adults were recruited into the China Kadoorie Biobank in 2004-2008 and medical records were retrieved on 20,299 first stroke cases with a 12-year follow-up. The study found that among 17,306 (85%) hospitalised cases with first ischemic stroke, anti-platelet treatment was used by 64%, lipid-lowering by 50% and blood pressure-lowering by 42% with positive trends by calendar year and hospital tier. In addition, among 2,623 (14%) with intra-cerebral haemorrhage, 53% used blood pressure-lowering treatment and 10% used lipid-lowering treatment. The study also showed that traditional Chinese Medicines were used 59% of ischemic stroke cases and 38% of intra-cerebral haemorrhage stroke cases. Overall, this study contained a huge amount of sample sizes and showed the contemporary use of evidence-based medical and surgical treatments for hospitalised cases of stroke in China. What is more, the study demonstrated that use of all evidence-based medical treatments increased substantially after the introduction of the nationwide universal healthcare system in China. The work is helpful for investigating the causes and consequences of variety of stroke management in clinical practice for stroke care in China. However, there remains some questions need to be answered to make the findings more valuable and meaningful for clinicians to determine the use of hospital treatments for stroke cases. Minor concerns:  1. The survey was conducted in 10 (5 urban, 5 rural) diverse areas in China, it is better to display what areas were included in the study, so the readers could understand whether the results were of generalization and representativeness. 2. It is better to present the information about when the hospital treatments including anti-platelet treatment, lipid-lowering treatment, and blood pressure lowering treatment were used in the recruited participants. 3. The study also estimated the rate of surgical treatments (Carotid endarterectomy or stenting) for any stroke, but few information was provided about when and where surgical treatments were performed in stroke cases. In which type of stroke cases?
---

4. The study demonstrated that intravenous thrombolytic treatment was administered to <1% of potentially eligible cases in the present study, which was pretty lower than that of 3.0-8.5% reported in the USA and 1.6% reported by Chinese National Stroke Registry (CNSR) in 2011. Could the authors explain the possible causes of the discrepancy?
5. The study showed that the overall prevalence of atrial fibrillation was very low, which was 0.3% in non-lacunar infarction (non-LACI) and 0.1% in lacunar infarction (LACI) cases. Did the routine ECG or holter was done in these cases?
6. In strengths part, page 4 line 1, the author described that “In a 12-year follow-up”, while in the introduction part, page 5 line 3, the authors described that “during an 11-year follow-up”. Please keep consistent.
7. Page 4 line 7, “which are 10-fold higher than those for ischemic heart disease” should be “which was 10-fold higher than those for ischemic heart disease”.

VERSION 1 – AUTHOR RESPONSE

Reviewer: 1.1

Dr. Shujuan Li, Zhejiang Pharmaceutical College Comments to the Author:

In this manuscript, using data from the China Kadoorie Biobank, the authors studied the current situation of hospital management and evidence-based treatments for stroke in Chinese hospitals. The results indicate that many evidence-based stroke therapies are still underused in China, while traditional Chinese medicine (TCM) has been widely used in the absence of sufficient efficacy and safety data. These results may provide evidence for stroke treatments and future studies. There are a few minor shortcomings to be ironed out before publication.

1. The time and sample size of this study have been described for many times in the article. Please pay attention to the consistency of the descriptions.

1) Page 3, lines 11-14: “The China Kadoorie Biobank recruited >0.5 million adults during 2004-08 from ten diverse areas and medical records were retrieved from 20,229 participants (n=261 hospitals) hospitalised with a first-ever incident stroke over an 11-year follow-up period.”

2) Page 4, lines 8-9: “In a 12-year follow-up of 0.5 million Chinese adults recruited into the China Kadoorie Biobank in 2004-2008”.

3) Page 6, line 24: “In brief, the 2004-8 baseline survey was conducted”.

4) Page 12, lines 3-5: “increased by calendar year of follow-up between 2005 and 2017.”

Response: Thank you for highlighting these discrepancies. CKB cohort is ongoing prospective study with regular data releases during follow up. The analysis in the present study were based on data release 17.0, which continued until 1 January 2018 and included a median duration of follow-up of 11 years. We have revised the paragraph on the strengths and limitations of the study to clarify the median duration of follow-up.

1.2. In the “Study population” section, the description of CKB is relatively detailed, but it is not clear about the population of the present study. It is suggested to add the first sentence of “Statistical analysis”(Page 9, lines 29-37) to the “Study population”, and the sentence should be amended to avoid ambiguity. Due to the large sample size of this study, it is suggested to add a flow chart of participants.

Response: We have moved the sentence into Study population and added a summary in the statistical analysis section.

1.3. In this article, silent cerebral infarction (SCI) was included in the classification of IS. Therefore, the description of IS and non-LACI needs to be adjusted.

Response: Thank you this helpful suggestion. Given that almost all silent infarction cases had lacunar infarcts, we have now revised the terminology for Silent LACI in both the text (page 7) and Tables 2 and 3.

1.4. "Surgical treatments" was mentioned in the "Introduction", but they are only mentioned again in the results and Table 2, and these results are not discussed in the "Discussion" section. Therefore, I suggest that this part can be deleted as appropriate.

Response: Done. We have now deleted use of surgical treatment for IS cases from Table 2.

1.5. Supplement the description of "Unajudicated stroke (N=26,965)" in the eTable 1. There is no description of this number in the manuscript or annotations.

Response: The present study did not include any unadjudicated cases. Hence, eTable 1 has now been removed.

Reviewer: 2

Dr. Jing Ding, Zhong Shan Hospital

Comments to the Author:

The study aimed to investigate the use of hospital treatments for stroke types in Chinese adults. Over 0.5 Million Chinese adults were recruited into the China Kadoorie Biobank in 2004-2008 and medical records were retrieved on 20,299 first stroke cases with a 12-year follow-up. The study found that among 17,306 (85%) hospitalised cases with first ischemic stroke, anti-platelet treatment was used by 64%, lipid-lowering by 50% and blood pressure-lowering by 42% with positive trends by calendar year and hospital tier. In addition, among 2,623 (14%) with intra-cerebral haemorrhage, 53% used blood pressure-lowering treatment and 10% used lipid-lowering treatment. The study also showed that traditional Chinese Medicines were used 59% of ischemic stroke cases and 38% of intra-cerebral haemorrhage stroke cases. Overall, this study contained a huge amount of sample sizes and showed the contemporary use of evidence-based medical and surgical treatments for hospitalised cases of stroke in China. What is more, the study demonstrated that use of all evidence-based medical treatments increased substantially after the introduction of the nationwide universal healthcare system in China. The work is helpful for investigating the causes and consequences of variety of stroke management in clinical practice for stroke care in China. However, there remains some questions need to be answered to make the findings more valuable and meaningful for clinicians to determine the use of hospital treatments for stroke cases.

Minor concerns:

2.1 The survey was conducted in 10 (5 urban, 5 rural) diverse areas in China, it is better to display what areas were included in the study, so the readers could understand whether the results were of generalization and representativeness.

Response: Thank you for these helpful comments. We have added the names of the 10 CKB regions to the text (page 5).

2.2 It is better to present the information about when the hospital treatments including anti-platelet treatment, lipid-lowering treatment, and blood pressure lowering treatment were used in the recruited participants.

Response: The present study was based on data from medical records including initial page, discharge summary, first admission record, and diagnostic tests (including CT and/or MRI test reports) of the first hospitalised events. Given this is not registry study and information on dates of such treatment were not always available from medical records (especially those from rural hospitals), we are unable to provide details on timing of such treatments for all cases.

2.3 The study also estimated the rate of surgical treatments (Carotid endarterectomy or stenting) for any stroke, but few information was provided about when and where surgical treatments were performed in stroke cases. In which type of stroke cases?

Response: For the same reason as outlined in response to Reviewer 2.2, we were unable to provide data on the timing of any surgical procedures.

2.4 The study demonstrated that intravenous thrombolytic treatment was administered to <1% of potentially eligible cases in the present study, which was pretty lower than that of 3.0-8.5% reported in the USA and 1.6% reported by Chinese National Stroke Registry (CNSR) in 2011. Could the authors explain the possible causes of the discrepancy?

Response: Thank you for these helpful suggestions. CNSR study was based on Tier 2 and 3 hospitals in urban areas only. The intravenous thrombolytic treatment was unlikely to be administered in tier 1 hospitals in rural areas. The present study consisted of 21% of tier 1 hospitals. We have acknowledged these limitations in the Discussion section (page 13).

2.5 The study showed that the overall prevalence of atrial fibrillation was very low, which was 0.3% in non-lacunar infarction (non-LACI) and 0.1% in lacunar infarction (LACI) cases. Did the routine ECG or holter was done in these cases?

Response: Thank you for these comments. As explained in response to Reviewer 2.5 the present study included 21% of tier 1 hospitals. Although routine ECGs were recorded in 88% of all first hospitalisations, Holter monitoring was conducted in only 7% of stroke cases. We have discussed these limitations in the Discussion section (page 13).

2.6 In strengths part, page 4 line 1, the author described that “In a 12-year follow-up”, while in the introduction part, page 5 line 3, the authors described that “during an 11-year follow-up”. Please keep consistent.

Response: Thank you for pointing out these discrepancies. We have now revised the paragraph on strengths and limitation (see page 3) and corrected the description of duration of follow up (see page 9).

2.7. Page 4 line 7, “which are 10-fold higher than those for ischemic heart disease” should be “which was 10-fold higher than those for ischemic heart disease”.

Response: Agree. Done (see page 4).

COI statements

Reviewer: 1

Competing interests of Reviewer: There are no conflicts of interest.

Reviewer: 2

Competing interests of Reviewer: No.

VERSION 2 – REVIEW

REVIEWER	Li, Shujuan Capital Medical University, beijing chaoyang hospital Department of neurology
REVIEW RETURNED	19-Oct-2021
GENERAL COMMENTS	No further comments.